# Fluorescent sensors reporting the activity of ammonium transceptors in live cells

Roberto De Michele[1,2], Cindy Ast[1,3], Dominique Loqué[4], Cheng-Hsun Ho[1], Susana LA Andrade[5], Viviane Lanquar[1], Guido Grossmann[1†], Sören Gehne[6], Michael U Kumke[6], Wolf B Frommer[1]*

[1]Department of Plant Biology, Carnegie Institution for Science, Stanford, United States; [2]Institute of Plant Genetics, Italian National Research Council (CNR-IGV), Palermo, Italy; [3]NanoPolyPhotonik, Fraunhofer Institute for Applied Polymer Research, Potsdam-Golm, Germany; [4]Feedstocks Division, Joint BioEnergy Institute, Emeryville, United States; [5]Department of Biochemistry, Institute for Organic Chemistry and Biochemistry, and BIOSS Center for Biological Signalling Studies, University of Freiburg, Freiburg, Germany; [6]Department of Physical Chemistry, Institute of Chemistry, University of Potsdam, Potsdam, Germany

**Abstract** Ammonium serves as key nitrogen source and metabolic intermediate, yet excess causes toxicity. Ammonium uptake is mediated by ammonium transporters, whose regulation is poorly understood. While transport can easily be characterized in heterologous systems, measuring transporter activity in vivo remains challenging. Here we developed a simple assay for monitoring activity in vivo by inserting circularly-permutated GFP into conformation-sensitive positions of two plant and one yeast ammonium transceptors ('AmTrac' and 'MepTrac'). Addition of ammonium to yeast cells expressing the sensors triggered concentration-dependent fluorescence intensity (FI) changes that strictly correlated with the activity of the transporter. Fluorescence-based activity sensors present a novel technology for monitoring the interaction of the transporters with their substrates, the activity of transporters and their regulation in vivo, which is particularly valuable in the context of analytes for which no radiotracers exist, as well as for cell-specific and subcellular transport processes that are otherwise difficult to track.

*For correspondence: wfrommer@stanford.edu

†Present address: Centre for Organismal Studies/Cell Networks, Ruprecht-Karls-Universität Heidelberg, Heidelberg, Germany

Competing interests: The authors declare that no competing interests exist.

## Introduction

Transport proteins play critical roles in cellular uptake and release as well as subcellular distribution of ions and metabolites. Due to their hydrophobic nature the characterization of transporters is challenging, and activity measurements typically depend on the use of radiotracers and heterologous expression systems. One of the major breakthroughs for characterizing channels has been the development of electrophysiology tools, enabling characterization of electrogenic transport in live cells (*Neher and Sakmann, 1992*). However, patch clamping has largely been limited to the analysis of processes at the cell membrane, or cells at the surface of sliced tissues. A new level of insight was provided by fluorescent proteins enabling us to observe cellular processes in living cells of intact tissues (*Tsien, 2006*). Fluorescence Resonance Energy Transfer (FRET) sensors have allowed us to monitor analyte levels and dynamics in live cells with minimal invasive methods (*Okumoto et al., 2012*). Particular FRET sensors can even be used to monitor biophysical processes such as membrane potential or tension between molecules in live cells, as well as the conversion of substrates of enzyme reactions by proteases and kinases. Yet to date, we have not been able to directly follow the activity of transporters and to monitor their regulation in vivo.

We rationalized that it may be possible to create transport activity sensors by monitoring structural rearrangements in a transporter with the help of environmentally sensitive genetically encoded

**eLife digest** Ammonium provides a vital source of nitrogen for bacteria, fungi and plants, and is produced by animals as a waste product of metabolism. High levels of ammonium can be toxic, so all organisms need to control their uptake or excretion of this substance. Ammonium transporters, which are highly conserved from bacteria to plants to humans, are essential for this process but, along with transporters in general, they are hard to study. Their activity can be examined in vitro by expressing them in heterologous systems—that is, in cells other than those in which they are naturally found. But in vivo studies must rely on indirect techniques such as monitoring radioactive isotopes or membrane potentials, and these cannot distinguish between the activity of ammonium transporters and uptake of ammonium through other routes.

One approach that has been successful in other fields is the use of fluorescent proteins that can signal conformational changes—such as those that occur when a transporter is activated—by a shift in fluorescence. Green fluorescent protein (GFP) is a commonly used fluorescent indicator, and a particularly useful variant is 'circularly permutated GFP'. This is GFP in which parts of the amino acid sequence have been rearranged without fundamentally changing the overall structure or function of the protein. Circularly permutated GFP can be fused to another protein in such a way that a conformational change in the second protein triggers a change in fluorescence that can be detected by fluorescence spectroscopy or microscopy.

Now, De Michele et al. have applied this approach to the study of both plant and yeast ammonium transporters. They constructed a library of fusion proteins made up of circularly permutated GFP and an ammonium transporter from the plant *Arabidopsis thaliana*—and found one version that functioned normally as a transporter but also produced a detectable change in fluorescence that correlated precisely with transporter activity.

De Michele et al. then used the same method to produce fluorescent indicator fusion proteins of two more ammonium transporters—a second isoform from *Arabidopsis* and one from yeast. These fluorescent sensors should be a great boon to researchers studying the ammonium transport system. Moreover, this approach could in theory be applied to other transporter proteins that are currently difficult to study, and so could help to open up research into a variety of transport processes.

fluorophores. Based on the pioneering work of *Miyawaki et al. (1997)*, FRET can be used to probe conformational rearrangements and has widely been exploited to develop genetically encoded sensors for small molecules. An alternative, again developed by Tsien's lab (*Baird et al., 1999*), is the use of circularly permutated GFP variants as reporters for conformational rearrangements, which have been exploited to create sensors for small molecules such as calcium, zinc and maltose (*Baird et al., 1999*; *Marvin et al., 2011*). Circularly-permutated versions of GFP (cpGFP) have successfully been used to monitor structural rearrangements of calmodulin (*Baird et al., 1999*; *Nagai et al., 2001*; *Nakai et al., 2001*; *Wang et al., 2008*; *Akerboom et al., 2009*). cpGFP is a protein in which the β-barrel structure has been rearranged in a way that ligand-induced conformational changes can influence the chromophore environment and consequentially the fluorescent properties of a chimeric sensor. Depending on pH, GFP and its variants have two excitation maxima ($\lambda_{ex} \sim 395$ and $475$ nm) (*Tsien, 1998*). Protein-cpGFP fusions, in which structural rearrangements affect the state of the chromophore as well as the hydrogen bonds between the chromophore and the surrounding β-barrel, can in principle be exploited to develop sensors that monitor transporter activity. Here we tested the hypothesis that ammonium transporters can effectively report transporter activity in live cells when fused to cpGFP.

Ammonium transport is key to nitrogen nutrition of bacteria, fungi and plants. Although mammals are unable to assimilate ammonium into amino acids, ammonium transporters play key roles in renal ammonium secretion and male fertility (*Biver et al., 2008*). Transporters for ammonium are conserved in bacterial, fungal, plant and animal genomes. Ammonium transport is electrogenic and is mediated by high affinity ammonium transporters of the AMT/MEP/Rhesus protein superfamily (*Marini et al., 1994*; *Ninnemann et al., 1994*; *Tremblay and Hallenbeck, 2009*). The genome of the plant Arabidopsis contains six AMT paralogs, four of which (AMT1;1, 1;2, 1;3, and 1;5) are partially redundant and together essential for ammonium acquisition (*Yuan et al., 2007*).

In plants, ammonium acquisition is highly regulated and is subject to feedback inhibition, potentially as a means of preventing accumulation of ammonium to toxic levels (*Wang et al., 1993*; *Kronzucker et al., 2001*). Recent results show that ammonium-triggered phosphorylation of a critical threonine in the cytosolic C-terminus of AMT1;1 leads to inhibition of ammonium transport by allosteric regulation in the trimeric transporter complex (*Loqué et al., 2007*; *Lanquar et al., 2009*). The close homolog AMT1;3, also functions as an electrogenic high-affinity ammonium transporter involved in nitrogen uptake in Arabidopsis roots (*Gazzarrini et al., 1999*). AMT1;3 functions in homo- and hetero-trimeric complexes with the coexpressed AMT1;1 and is allosterically regulated (*Yuan et al., 2013*). AMT1;3 not only functions as a transporter, but in addition controls root architecture by a process similar to the one found for the yeast MEP2 transceptor (dual function transporter and receptor), a protein that mediates ammonium transport and regulates pseudohyphal growth (*Boeckstaens et al., 2007*; *Lima et al., 2010*). The term transceptor had been introduced by Thevelein to describe the dual function of the general amino acid permease Gap1p or the phosphate transporter Pho84p in transport and signaling (*Kriel et al., 2011*).

Most of the progress in the ammonium transport field has been made with the help of yeast and *Xenopus laevis* oocytes as heterologous expression systems. By contrast, in planta studies mostly rely on the use of either stable isotopes ($^{15}$N), or radiotracer analogs ($^{14}$C-methylammonium). Monitoring ammonium-induced depolarization of membrane potential has successfully been deployed in plants (*Wang et al., 1994*), yet it can not differentiate between transport mediated by individual AMTs and non-selective cation channels. Therefore, it has not been possible to identify the networks that regulate the substrate-triggered feedback inhibition of ammonium uptake in plant roots. Thus there is a need for new tools that enable us to monitor transport activity in intact root tissues.

Here we created an activity sensor by inserting cpGFP into a cytosolic loop of the Arabidopsis ammonium transporter AMT1;3. Thorough characterization demonstrates that the chimera reports processes that occur in the transport cycle, although at present it is not possible to differentiate substrate binding or steps in the transport cycle. We project that this approach is transferable to other proteins, a notion supported by the successful creation of additional sensors for the Arabidopsis paralog AMT1;2 as well as the distantly related yeast ammonium transceptor MEP2.

## Results

### Identification of suitable fluorophore insertion sites

To engineer a transporter that reports substrate-dependent changes in conformation, we inserted fluorescent proteins (FP) into *Arabidopsis thaliana* AMT1;3 (*Figure 1*). Monomeric teal fluorescent protein (mTFP), Venus or a modified circularly permutated GFP (mcpGFP), which had successfully been employed to create highly sensitive, single-fluorophore sensors for calcium and maltose (*Tian et al., 2009*; *Marvin et al., 2011*), were inserted into intracellular loops of AMT1;3. The functionality of the transporter fusions was tested by complementation of ammonium uptake deficiency in a yeast mutant lacking endogenous ammonium transporters (*Marini et al., 1997*). AMT1;3 activity was extremely sensitive to manipulation of loops 7–8 and 9–10 or the cytosolic C-terminus (*Figure 1B*). However, modification of loop 5–6 (L5–6, position K233) by insertion of either two amino acid residues (encoded as part of the restriction site *XbaI*) or mcpGFP was tolerated (*Figure 1B*). In contrast, mTFP or Venus insertions inhibited transport activity. Interestingly, L5–6 is located between the two pseudo-symmetric halves of the protein and connects two transmembrane helices (TMH-V and -VI) that contain residues postulated to be directly involved in recruitment, gating and substrate translocation (*Figure 1C*; *Andrade et al., 2005*). Moreover, TMH-V has been proposed to oscillate during substrate transport (*Andrade et al., 2005*; *Inwood et al., 2009*). L5–6 thus may be a prime location for detecting structural rearrangements through readout of a conformation-sensitive FP.

### Generation of the ammonium sensor AmTrac

In initial tests, addition of ammonium to yeast cells expressing the AMT1;3-mcpGFP fusion did not lead to detectable changes in fluorescence intensity (FI; data not shown). However, it is known that the cpGFP-based sensors for calcium and maltose are exquisitely sensitive to modifications in the linker sequences connecting to the respective binding proteins (*Wang et al., 2008*; *Akerboom et al., 2009*; *Marvin et al., 2011*). To explore whether linker modification would affect the ability of the chimera to detect structural rearrangements of AMT1;3 during transport, we created 24 variants by altering the

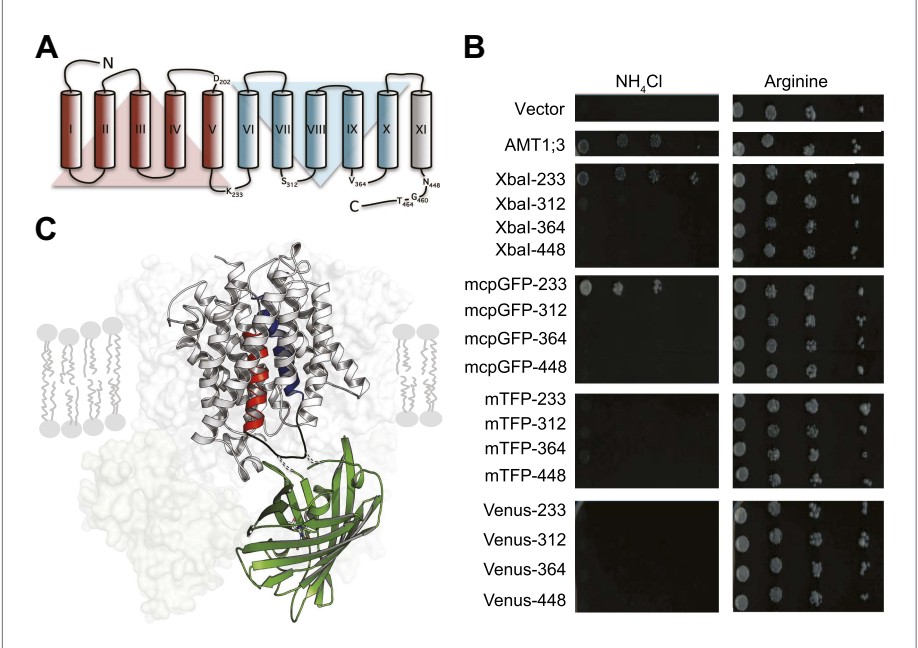

Figure 1. Design of fusion constructs. (**A**) Topological representation of AMT1;3 by HMMTOP (**Tusnady and Simon, 2001**). Eleven TMH are organized in a pseudo-symmetric structure (TMH I-V and TMH VI-X) with an extra terminal TMH-XI that directs the C-terminus to the cytosol. The position of the residues preceding the insertion points of FPs in L5–6, 7–8, 9–10 and in the C-tail are indicated. Residues D202, G460 and T464, important for the activity of the transporter, are also shown. (**B**) The functionality of the transporters was measured as growth of the yeast Δ*mep1,2,3* mutant transformed with AMT1;3-FP fusions and grown on solid media containing 2 mM NH₄Cl or 1 mM arginine (growth control) as the sole nitrogen source for three days. Numbers indicate the position in AMT1;3 preceding the insertion site. Vector: empty vector served as the negative control. (**C**) Three-dimensional model of one AMT-mcpGFP chimeric protein based on the crystal structures of Af-AMT1 (2B2H) and cpGFP (3evp). One monomer is shown in cartoon and the rest of the trimeric complex is represented as a shaded surface in the background. mcpGFP (green) was inserted in position 233 of L5–6 of AMT1;3, connecting TMH-V and -VI (red and blue, respectively).

linkers connecting AMT1;3 and mcpGFP, exploiting information generated during optimization of calcium and maltose sensors (**Wang et al., 2008**; **Akerboom et al., 2009**; **Marvin et al., 2011**). All 24 variants maintained transport activity (**Figure 2A**); importantly some variants showed a FI response to addition of 1 mM NH₄Cl (**Figure 2B**). The variant showing the strongest response (40% FI change) was named AmTrac (Ammonium Transporter coupled to mcpGFP). In AmTrac, mcpGFP had been inserted into position 233 of AMT1;3, centrally located in the cytosolic loop L5–6 and flanked via linker sequences containing Leu-Glu and Phe-Asn at the N- and C-terminus, respectively (**Figure 2C,D**). To explore the tolerance of L5–6 of AMT1;3 to mcpGFP insertion, we varied the insertion position in the loop (**Figure 3A**). Both transport activity and fluorescence response were highest in variants with mcpGFP inserted in central positions of L5–6, with a clear correlation between activity and FI change (**Figure 3B**). Loop length also proved to be critical. In variants with incremental deletions in the middle of L5–6 around the insertion point of mcpGFP, deletion of four or more residues abolished transport activity and the fluorescence response (**Figure 3C**). Importantly, all variants that responded to ammonium with a FI change were functional transporters, indicating that AmTrac reports transport activity. Fluorescent variants that were able to mediate transport but showed no FI change (such as AMT1;3-mcpGFP-233) were used as control sensors to exclude potential effects of other parameters such as intracellular ammonium, proton accumulation or depolarization on the FI change.

## Kinetics, selectivity and reversibility of AmTrac

Initial experiments showed that the FI response of AmTrac was proportional to the ammonium concentration added to the yeast cells (**Figure 4A**). The mcpGFP insertion into AMT1;3 did not significantly affect the affinity constant for ammonium transport ($K_m \sim 50$ μM) when tested in *Xenopus* oocytes

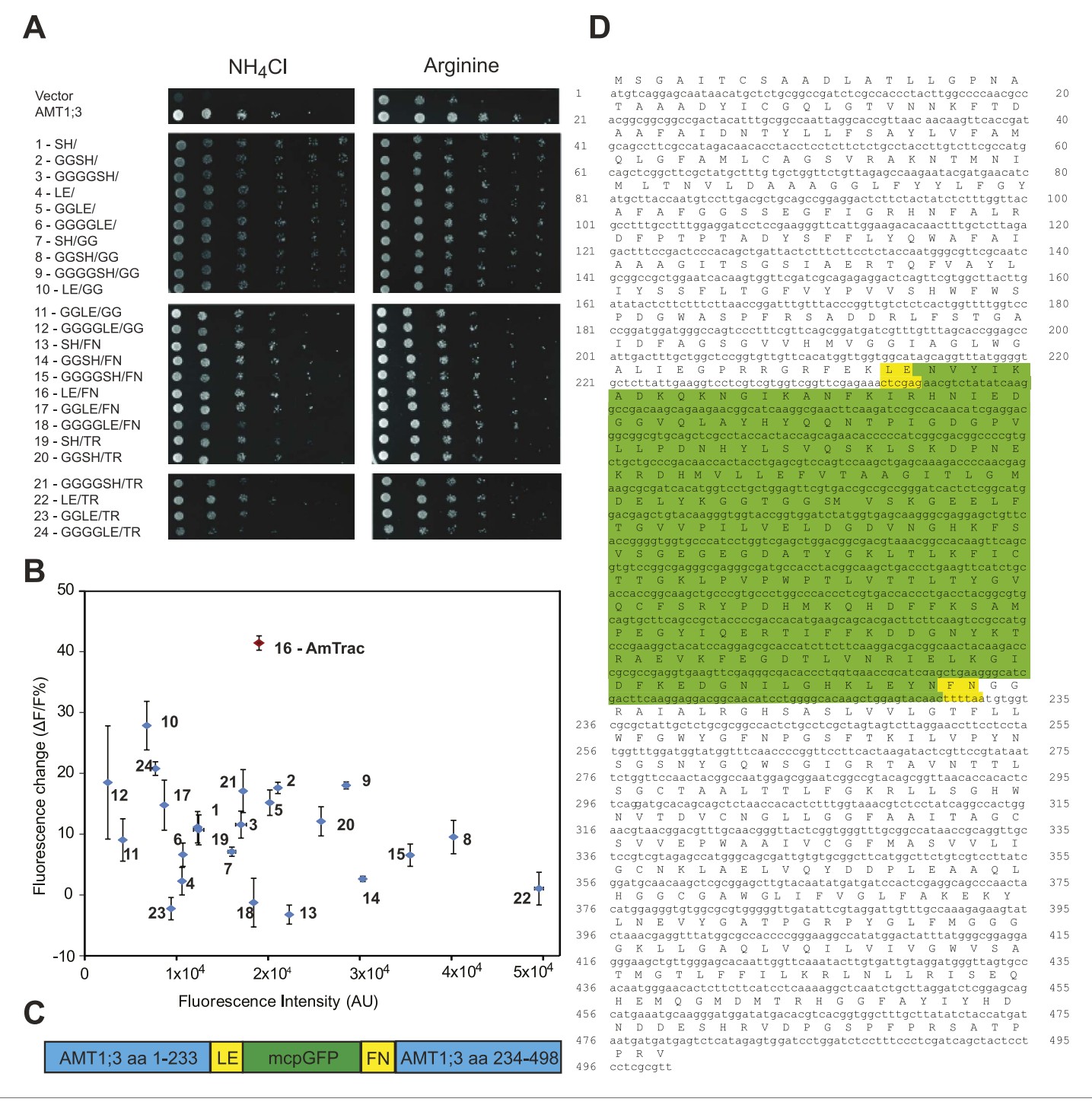

**Figure 2**. Development of AmTrac. (**A**) Growth of the yeast Δ*mep1,2,3* mutant transformed with fusion variants on solid media containing 2 mM NH₄Cl or 1 mM arginine (growth control) as the sole nitrogen source for 3 days. Composition of linkers connecting AMT1;3 and mcpGFP are indicated. Linkers at the N- and C-termini of mcpGFP are indicated in letter code and separated by a slash. In the cases of variants 1–6, no linkers were inserted between the C-terminal sequence of mcpGFP and the second part of AMT1;3. (**B**) Screen of 24 linker variants for fluorescence intensity before addition of ammonium and fluorescence intensity change after addition of 1 mM NH₄Cl (mean ± SE; n = 3). Variant 16 (in red), carrying LE/FN as linkers, named AmTrac, showed the highest change in fluorescence intensity. (**C**) Schematic representation of AmTrac. Linkers between AMT1;3 and mcpGFP are indicated in yellow. (**D**) Protein sequence of AmTrac. Residues in 'yellow' constitute synthetic linker segments. Residues in 'green' correspond to the mcpGFP moiety. Numbers indicate amino acid position in AMT1;3.

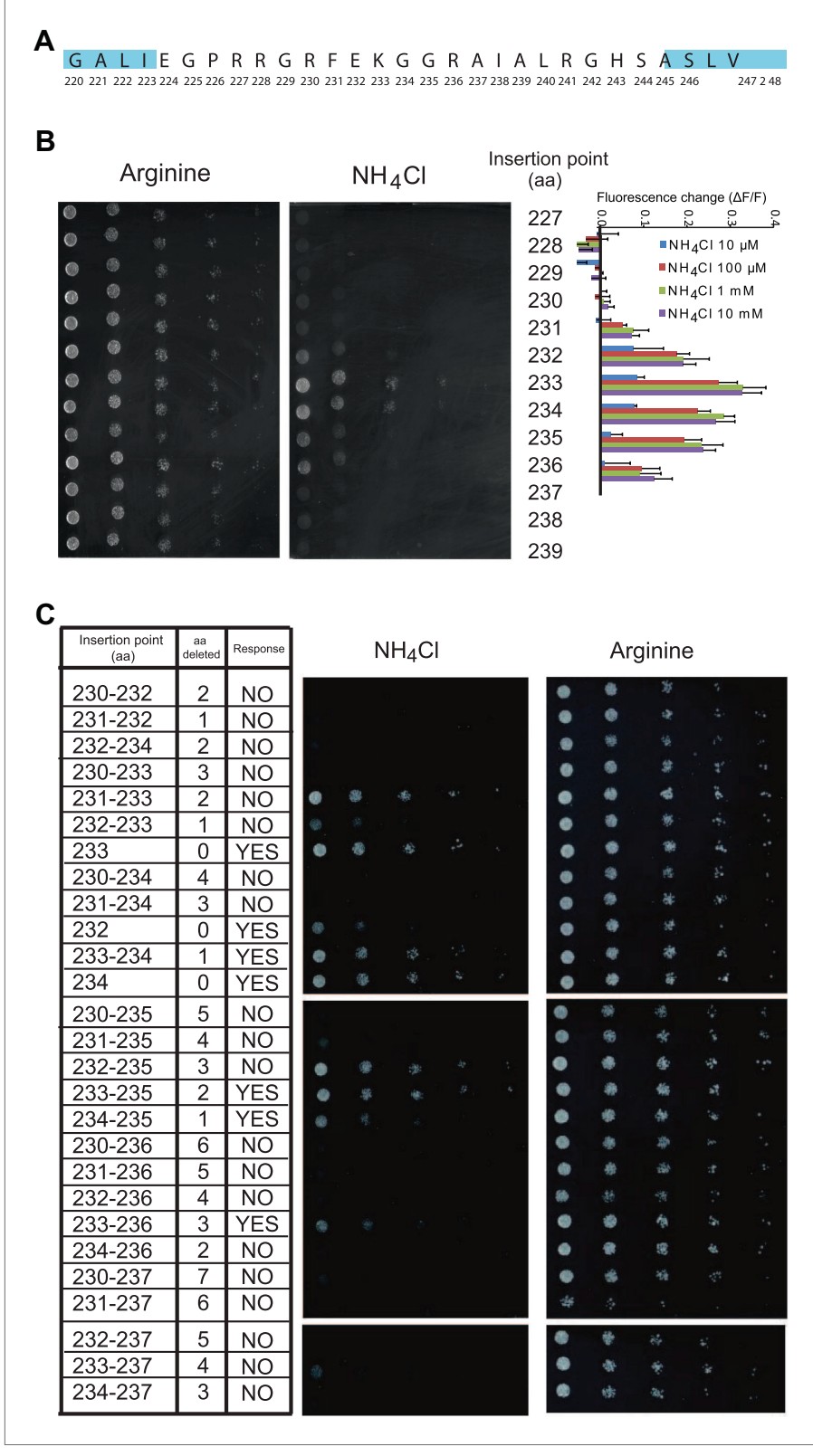

**Figure 3**. Effect of L5 on transport and fluorescence response. (**A**) Amino acid sequence of the L5–6 region of AMT1;3. Residues in 'blue' correspond to TMH 5 (left) and TMH 6 (right). (**B**) Influence of the insertion position of mcpGFP into L5 on transport activity and fluorescence response. Left panels show the growth assay of the yeast Δ*mep1,2,3* strain transformed with insertion variants on solid media containing 2 mM $NH_4Cl$ or 1 mM arginine (growth control) as the sole

*Figure 3. Continued on next page*

*Figure 3. Continued*

nitrogen source for 3 days. Numbers indicate the insertion site within AMT1;3 (residue preceding the point of insertion of mcpGFP). Right graph shows the fluorescence response of the variants to addition of the indicated concentrations of $NH_4Cl$. Data were normalized to the water-treated control (0; mean ± SE; n = 2). (**C**) Growth and response of AmTrac variants with deletions in L5–6. Growth was analyzed as described in *Figure 2A*. Numbers in the left column indicate the position of the insertion in AMT1;3; two numbers indicate residues preceding and following the mcpGFP insertion. Right column named "Response" indicates whether the corresponding variant responded to addition of 1 mM $NH_4Cl$ with a fluorescence change. The original AmTrac corresponds to insertion after aa 233.

using two-electrode voltage clamp (TEVC) analysis (*Figure 4B*). Notably, the affinity constant for the FI response of AmTrac to $NH_4Cl$ ($EC_{50}$ 55 ± 7 µM) is strikingly similar to that reported here in oocytes and to the reported transport kinetics of AMT1;3 as measured in roots of Arabidopsis plants expressing only AMT1;3 ($K_m$ 57 µM; *Figure 4C*; *Yuan et al., 2007*).

The saturation kinetics observed for the FI response can be interpreted as the dose-dependent conversion of an increasing number of transporters into a new state, for example from inactive to active or unbound to bound. We would assume that this process is reversible, so that AmTrac returns to the inactive or unbound state when ammonium is removed. To test the reversibility of the sensor response directly, we recorded the FI of single cells trapped in a microfluidic device and exposed to ammonium pulses. AmTrac responses were detectable in single cells, were concentration-dependent, and were readily reversible, demonstrating that AmTrac can be used in vivo to quantify ammonium transport activity and to observe kinetics of the chimeric transporter in response to substrate availability (*Figure 4D*).

The unique selectivity of AMT1 for ammonium over potassium was retained for FI responses (*Figure 4E*). Also other cations did not induce a change in FI. $NH_4NO_3$ and $(NH_4)_2SO_4$ elicited the same FI change as $NH_4Cl$, demonstrating that chloride does not affect the FI response. Furthermore, addition of equimolar concentrations of sodium, potassium, calcium, magnesium, manganese or zinc did not trigger a change in FI (*Figure 4E*). Together these data strongly support the hypothesis that AmTrac measures ammonium concentrations and/or reports conformational state changes of the transporter.

## Development of improved AmTrac variants

The FI of AmTrac was relatively low, limiting the dynamic range and the signal-to-noise ratio (SNR). To create sensors with enhanced SNR, we randomly substituted residues in the two-amino acid linker directly preceding mcpGFP. We identified AmTrac versions that maintained transport activity and showed high FI and a large ammonium-induced FI response. Interestingly, the majority of the brightest variants carried a Ser residue instead of a Glu in the position immediately preceding the mcpGFP insertion (*Figure 5A*). AmTrac-LS, a sensor with a Leu-Ser linker, was as responsive to ammonium as AmTrac (~40% FI change), but was approximately fivefold brighter (*Figure 5B*). Fluorescence excitation and emission spectra of intact yeast cells expressing the improved variant AmTrac-LS were recorded. The excitation spectra revealed two maxima, a minor peak of the protonated chromophore at $\lambda_{ex}$ ~ 380 nm and a major peak of the deprotonated chromophore at $\lambda_{ex}$ ~ 495 nm. For the emission spectra one maximum was detected at $\lambda_{em}$ ~ 513 nm (*Figure 5C*). The spectra are similar to those reported for unmodified EGFP. No shifts in the fluorescence maxima were observed, while FI decreased by 30–40% in a concentration-dependent manner when ammonium was added .The linker modifications had no significant effect on the transport kinetics (tested for AmTrac-LS; *Figure 4B*), nor on the $EC_{50}$ of the fluorescence response (shown for AmTrac-IS, Ile-Ser linker; *Figure 5D*). AmTrac-LS responses were monitored in single yeast cells as described above and also showed reversibility of the FI change when ammonium was removed (*Figure 5E*).

## Strict correlation between transport activity and fluorescence output

To further corroborate the correlation between transport activity and ammonium-induced FI change, mutations (D202N, G460D, T464D) known to inactivate AMT1 transporters were introduced into AmTrac (*Loqué et al., 2007*; *Neuhäuser et al., 2007*). While these mutations did not affect plasma membrane localization of AmTrac (*Figure 6A*), both transport activity and fluorescence response were abolished (*Figure 6B,C*), indicating that transport activity is tightly correlated with the FI response of AmTrac. Alternatively, accumulation of intracellular ammonium, and not its transport, could affect mcpGFP fluorescence. Since the *Δmep1,2,3* yeast strain lacks endogenous ammonium transporters,

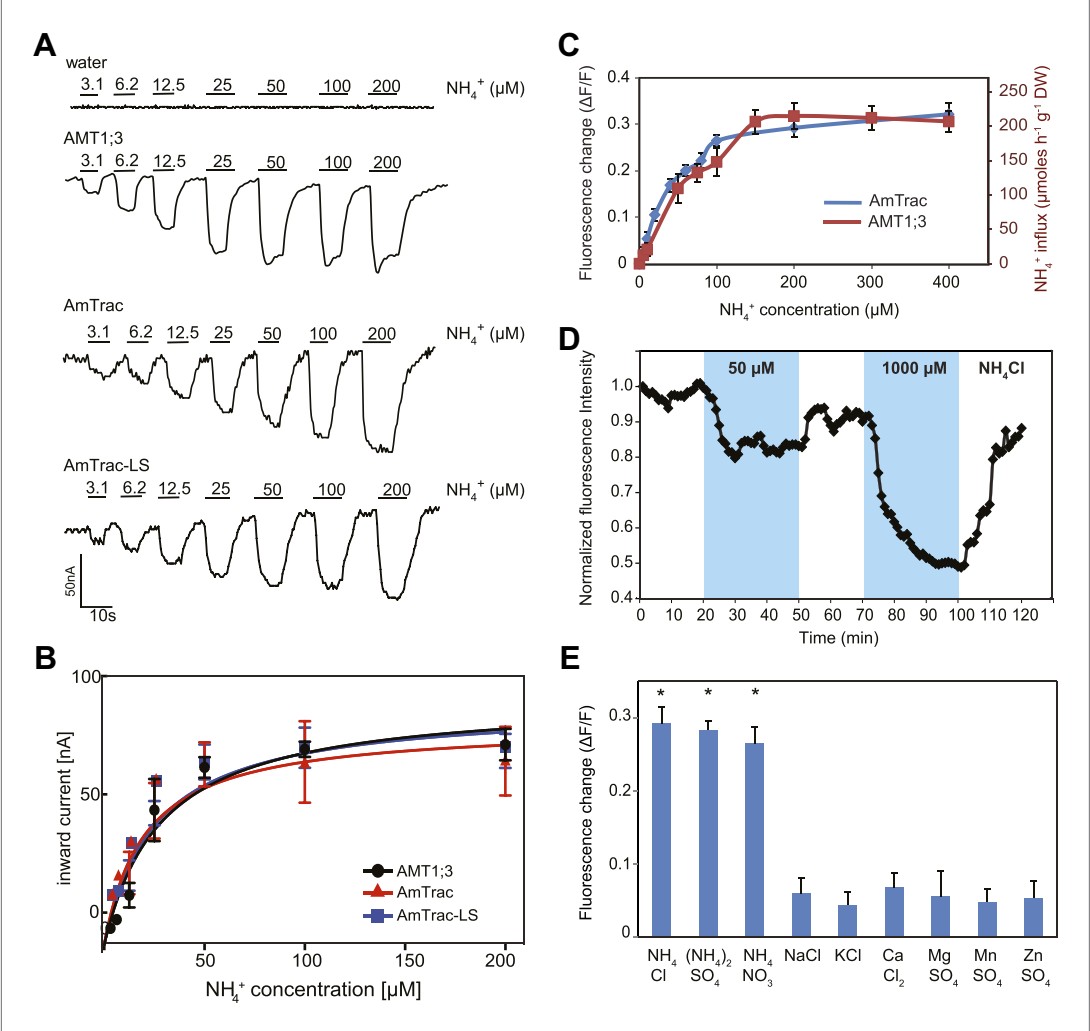

**Figure 4**. Characterization of AmTrac transport and responses. (**A**) Currents recorded in single oocytes injected with water, AMT1;3, AmTrac, or AmTrac-LS, and perfused with $NH_4Cl$ at the indicated concentrations. (**B**) Kinetics of $NH_4^+$-induced currents of AMT1;3, AmTrac, and AmTrac-LS. The $Kms$ were 55 ± 15 μM, 51 ± 24 μM, and 57 ± 19 μM, respectively. The data were fitted to Michaelis–Menten kinetics. Oocytes were clamped at −120 mV (independent data from three different oocytes recorded from three different frogs). (**C**) Titration of the fluorescence response of AmTrac in yeast (blue, left y-axis) and of ammonium uptake of AMT1;3 in plants (red, right y-axis; **Yuan et al., 2007**). Data are normalized to water-treated controls (0) (mean ± SE; n = 3). (**D**) Response of a single yeast cell expressing AmTrac to pulses of $NH_4Cl$ at the indicated concentrations (blue frames). (**E**) Substrate specificity. Yeast cells expressing the sensor were treated with the indicated salts at 1 mM concentration. Data are normalized to water-treated control (0) (mean ± SE; n = 3). Only data for the ammonium treatments were significantly different from control (SNK test: *p<0.01).

ammonium uptake in these cells depends on episomally expressed transporters. AmTrac mutants unable to transport ammonium also prevent intracellular ammonium accumulation. To discriminate between the hypotheses that AmTrac reports conformational states of the transporter or responds to accumulation of ammonium in the cytosol, AmTrac and AmTrac-D202N, -G460D, -T464D mutants were expressed in wild type yeast carrying endogenous ammonium transporters (MEPs; **Figure 6B**). In the presence of an independent set of functional ammonium transporters, none of the transport-deficient AmTrac versions produced an FI response (**Figure 6D**), providing strong evidence that cytosolic ammonium accumulation did not elicit the FI response. As a further test, generation of intracellular ammonium from arginine catabolism (**Marini et al., 1997**) failed to trigger an AmTrac response (**Figure 6E**). Taken together, these results indicate that AmTrac responds to extracellular and not intracellular ammonium levels. However our data do not exclude the possibility that other mutations can uncouple of the two processes, that is, to generate a protein that responds with a FI change to addition of ammonium but is unable to mediate transport across the membrane.

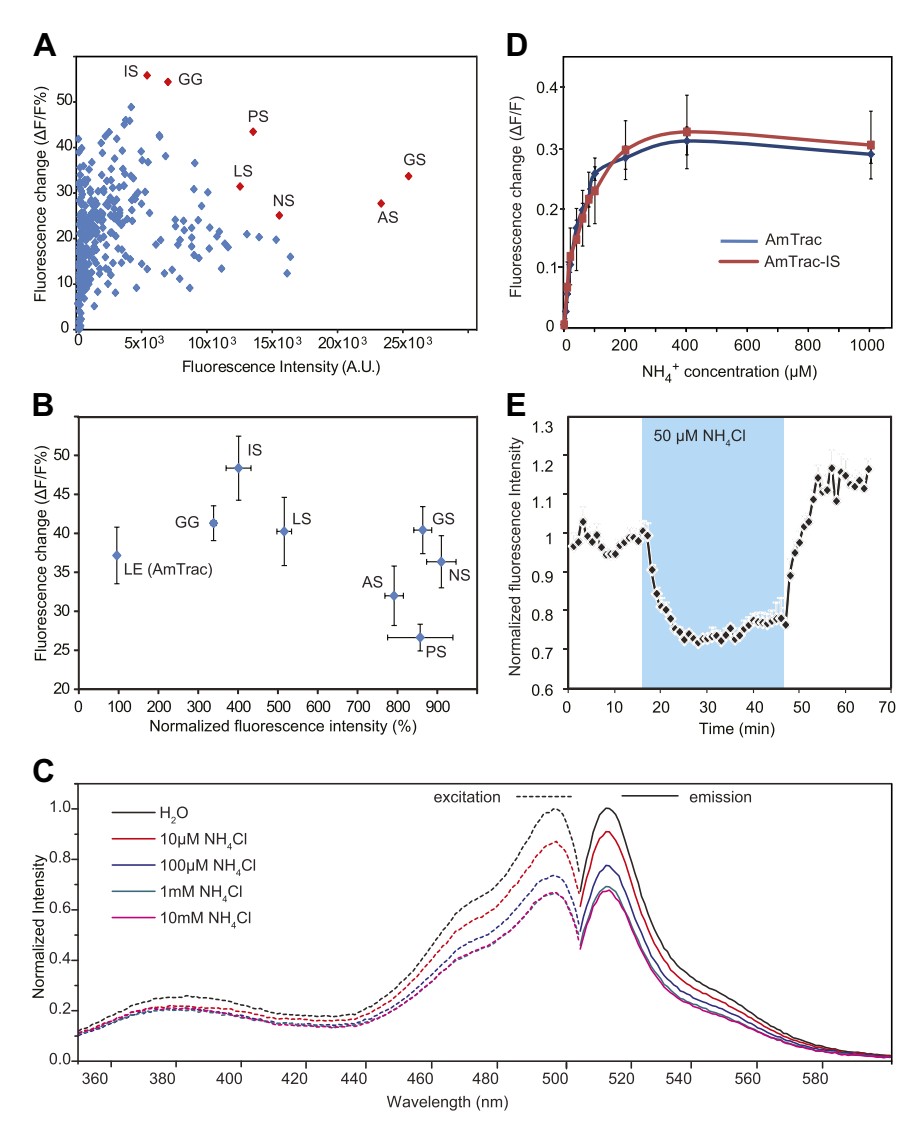

**Figure 5**. Development of improved variants of AmTrac. (**A**) Screen for improved sensor variants. Fluorescence intensity and fluorescence response of ~350 random variants of the C-terminal linker of mcpGFP to addition of 1 mM NH$_4$Cl. The brightest and most responsive variants (in red) were sequenced and composition of the linker (LE in AmTrac) is reported as letter code. Note that most variants carry a Ser residue in the last position of the linker. (**B**) Fluorescence intensity and fluorescence response after addition of 1 mM NH$_4$Cl, normalized to values of AmTrac (100%) of yeast expressing the best variants identified in (**A**) (mean ± SE; n = 3). (**C**) Steady-state fluorescence spectra of AmTrac-LS recorded at $\lambda_{exc}$ 485 nm and $\lambda_{em}$ 514 nm, respectively, with addition of NH$_4$Cl at the indicated concentrations, or water (as control). Fluorescence intensities were normalized to the major peak of the water control (=1). (**D**) Titration of the fluorescent response of AmTrac (blue line) and AmTrac-IS (red line). Data are normalized to water-treated controls (0; mean ± SE: n = 3). AmTrac kinetics shown here for comparison are the same as *Figure 4C*. (**E**) Single cell response of AmTrac-LS. Individual cells trapped in a microfluidic plate were perfused with 50 mM MES buffer pH 6.0, or a pulse of 50 µM NH$_4$Cl in buffer (blue box). Data were normalized to the initial value (mean ± SE; n =3).

## Reconstitution of output by restoration of transport in suppressor mutants

The allosteric trans-activation mechanism used by AMTs for feedback inhibition at elevated ammonium levels (*Loqué et al., 2007*; *Lanquar et al., 2009*) provided us with an opportunity to test whether restoration of transport activity in non-functional AmTrac mutants reconstitutes the FI response.

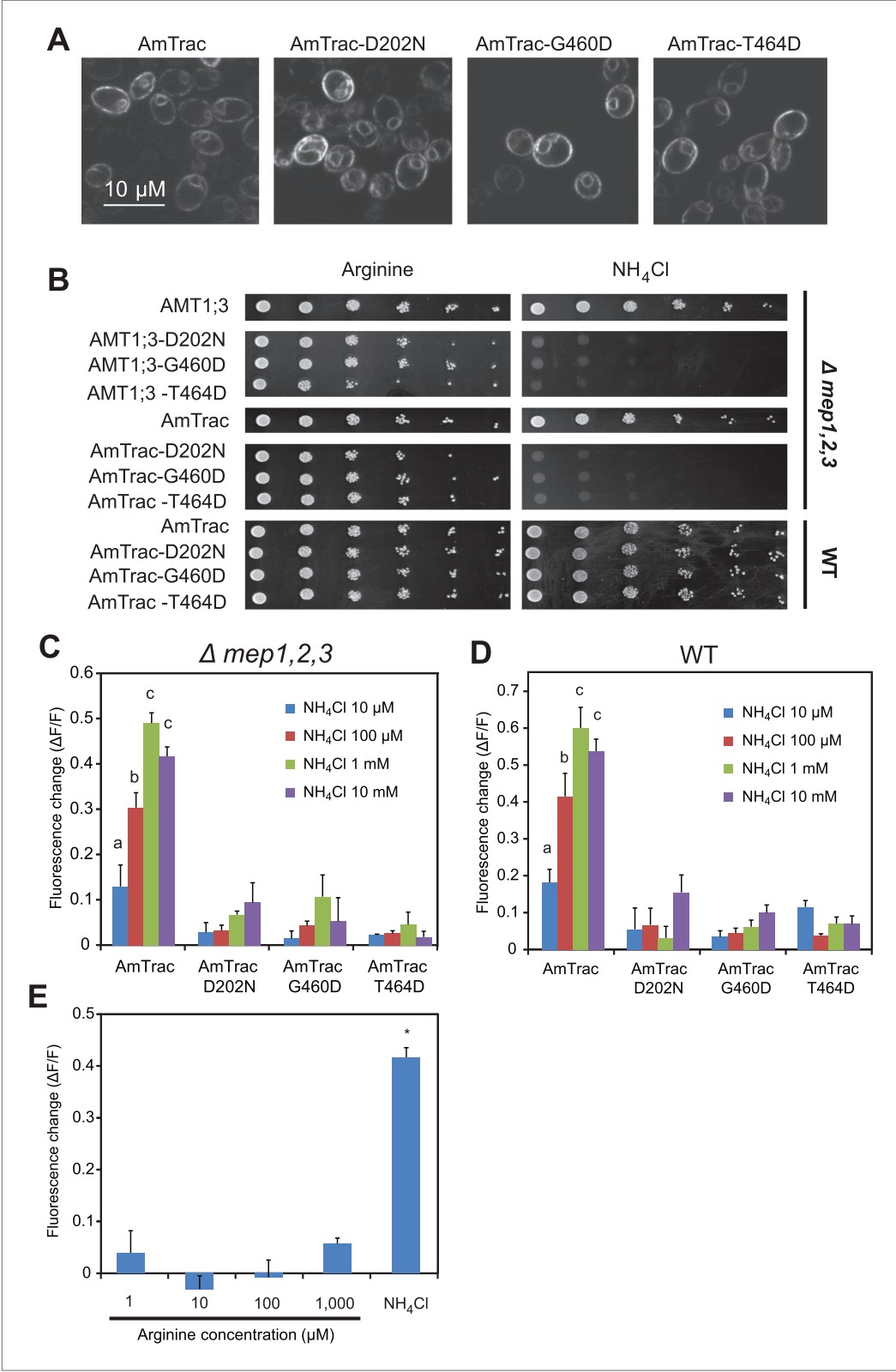

**Figure 6**. AmTrac mutant analysis. (**A**) Confocal z-section of yeast expressing AmTrac or its inactive variants carrying mutations D202N, G460D or T464D. Bar 10 μm. (**B**) Growth complementation of the *Δmep1,2,3* or wt yeast expressing AmTrac or its inactive variants on solid media containing 2 mM NH$_4$Cl or 1 mM arginine (growth control) as sole nitrogen source for 3 days. Endogenous MEPs in wt strain are not affected by expression of mutant variants. *Figure 6. Continued on next page*

*Figure 6. Continued*

(**C**) Fluorescence response of Δ*mep1,2,3* or (**D**) wt yeast expressing AmTrac or the transport-inactive variants D202N, G460D or T464D. Data were normalized to water-treated controls (0) (mean ± SE; n = 3). Only yeast cells expressing AmTrac showed significantly different responses (SNK test: p<0.01). (**E**) Response of AmTrac to addition of arginine. Fluorescence response of Δ*mep1,2,3* yeast cells expressing AmTrac treated with the indicated concentrations of arginine or 1 mM $NH_4Cl$. Data were normalized to water-treated controls (0) (mean ± SE; n = 3). Only the ammonium treatment was significantly different from control (SNK test: *p<0.01). Note that arginine addition to yeast cells has been shown to lead to increased cytosolic levels of ammonium (***Marini et al., 1994***).

Previous work had shown that the C-terminus of AMT1 acts as a trans-activation domain in the trimeric AMT1 complex (***Loqué et al., 2007***). Mutations affecting the cytosolic C-terminus block AMT1 activity, and activity can be restored by suppressor mutations in the cytosolic loops or mutations in the pore region (***Loqué et al., 2007***; ***Neuhäuser et al., 2007***). A saturating multicopy suppressor screen with the inactive mutant AmTrac-T464D (analogous to T460D in AMT1;1) (***Loqué et al., 2007***) identified eight gain-of-function mutations (***Figure 7A*** and ***Table 1***), seven in the pore region and one pseudo-reversion, D464V (***Figure 7B–D***). The extent to which the suppressors were able to restore transport activity, as measured by growth, was highly correlated with the FI response ($R^2$=0.72; ***Figure 7E,F***), further supporting the tight link between transport activity and FI response.

## Engineering of the low affinity sensor AmTrac-100μ

Interestingly, one of the suppressor mutants, AmTrac-T464D-A141E, mediated hypersensitivity to elevated ammonium concentrations (***Figure 7A***). Hypersensitivity, as shown in the case of the AtAMT1;1-T460D suppressor mutant Q57H (***Loqué et al., 2009***) can be caused by increased ammonium uptake capacity. TEVC analysis demonstrates that AmTrac-T464D-A141E functions as a low affinity/high capacity ammonium transporter with a $K_m$ ~ 3.5 mM (***Figure 8A***). The fluorescence response of AmTrac-T464D-A141E (AmTrac-100μ) also increased ($EC_{50}$ = 101 ± 10 μM; ***Figure 8B***). Once again, affinity and activity of the transporter correlated with the fluorescence change, suggesting a tight link between transport and fluorescence response.

## Single step construction of additional sensors: AmTrac1;2 and MepTrac

To test whether the strategy of creating activity state sensors by mcpFP insertion is generalizable, we inserted mcpGFP into the central loop of other ammonium transporters; that is, the paralog AMT1;2 from Arabidopsis and the transceptor MEP2 of *S. cerevisiae* (***Marini et al., 1997***). Phylogenetic analyses place AMT1;3 and AMT1;2 into the same subfamily. Yeast MEPs belong to a distantly related clade; and MEP2 and AMT1;3 share only 22% identity at the protein level. Notwithstanding, insertion of mcpGFP in the center of L5–6 of both AMT1;2 and MEP2 (after amino acids R242 and R217, respectively) with the same linkers as in AmTrac-LS, generated two functional sensors (named AmTrac1;2 and MepTrac) that maintained transport activity (***Figure 9A***) and responded to ammonium addition with a fluorescence change (***Figure 9B***). The original MepTrac showed only a low response when tested at pH 6. In contrast to AMTs, MEP2 has a pH optimum at low pH. Histidine-194 in the pore of MEP2 has been shown to be responsible for the lower pH optimum (***Boeckstaens et al., 2008***). A variant of MepTrac carrying the H194E mutation showed a significantly increased FI response, in agreement with the broader pH optimum conferred by this mutation (***Boeckstaens et al., 2008***; ***Figure 9B***). Similar to AmTrac, the response of AmTrac1;2, MepTrac and MepTrac-H194E to ammonium addition were dose-dependent. It is noteworthy that the AmTrac1;2 and MepTrac sensors were generated in a single step by using the linker and position information acquired during AmTrac development. We hypothesize that based on these results it will be possible to convert other transporters with similar twofold pseudo-symmetry into activity state sensors as well (***Forrest, 2013***).

## Discussion

Genetically encoded fluorescent biosensors are powerful tools for characterizing in vivo processes involved in cell signaling and physiology. They offer new tools for genetic screens, for example, for identifying regulatory networks, or they can be used for screening chemical libraries to identify new drugs (***Bermejo et al., 2013***). The existing tools, that is, FRET sensors for small molecules allow us to monitor steady state levels and dynamic changes of analytes such as ions, signaling compounds,

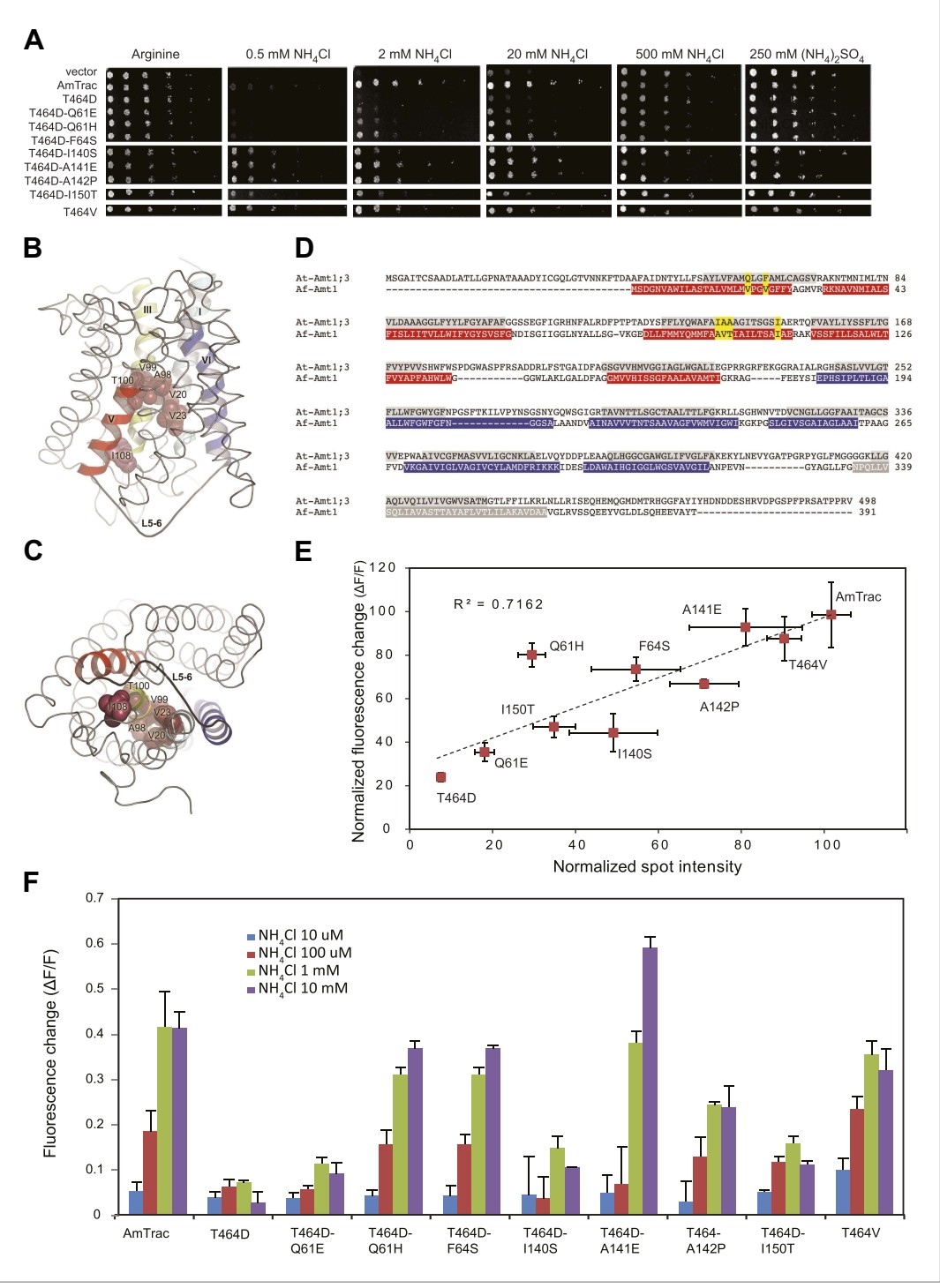

**Figure 7**. Suppressor mutants restore transport and fluorescence response. (**A**) Growth of the yeast Δ*mep1,2,3* strain transformed with suppressor mutants and grown on solid media containing the indicated concentrations of NH₄Cl, (NH₄)₂SO₄ (as anion control) or 1 mM arginine (growth control) as the sole nitrogen source for 3 days. Note that yeast expressing AmTrac-T464D-A141E grew poorly at high concentrations of ammonium, suggesting high capacity transport activity leading to ammonium toxicity. (**B**) Lateral view and (**C**) cytoplasmic side view of AfAmt1 according to the crystal structure (***Andrade et al., 2005***). The corresponding residues in AfAmt1 that suppress the T464D mutation in AmTrac are indicated by spheres. TMH-V is shown as red helix, TMH-VI in blue. The connecting L5–6 is labeled. Note that residues corresponding to cis-suppressing mutations reside in the internal pore region.
*Figure 7. Continued on next page*

*Figure 7. Continued*

(**D**) Sequence alignment of AMT1;3 from Arabidopsis (At-Amt1;3) and Af-Amt1 from *A. fulgidus*. The residues belonging to TMH domains of the two pseudo-symmetric halves of AfAMT1 are shown in red and blue. C-terminal TMH-XI of AfAMT1 is shown in grey, with white font. Predicted TMHs of AMT1;3 are highlighted in grey, with black font. The corresponding residues identified in the suppressor screen of AmTrac-T464D are indicated in both sequences (yellow). (**E**) Correlation between transport efficiency (growth in 2 mM NH$_4$Cl) and fluorescence change after addition of 1 mM NH$_4$Cl of the suppressor mutants. Data are normalized to values obtained for AmTrac (=100) (mean ± SD; n = 3). (**F**) Fluorescence response of suppressors to addition of the indicated concentrations of NH$_4$Cl. Data were normalized to water-treated controls (0) (mean ± SE; n = 3).

metabolites and substrates of proteases (*Okumoto et al., 2012*). Combined with information on the cellular and subcellular localization of receptors, enzymes and transporters, they can be used to obtain a first picture of the signaling and metabolic networks. However, tools for measuring the activity of transporters in vivo have been limited to electrophysiology, which can not differentiate between transporter isoforms, and is difficult to use for measuring transporter activity in cells in live tissues or even subcellular compartments. Here we developed a prototype for such activity sensors using the important class of ammonium transporters conserved from archaebacteria to higher plants and metazoa, including the human Rhesus factors. We made use of the effect of conformational rearrangements in AMTs/MEPs transceptors on the fluorescence emitted by circularly permuted mcpGFPs that have been inserted into an intracellular loop of the transporters. These fluorescence-based sensors report the activity of the transporter in vivo and can be used to quantify the activity of a specific transporter in vivo. Information on the activity of transporters is relevant for determining the regulation of individual isoforms in live cells, to identify and characterize regulators, to monitor transport in cells otherwise inaccessible to transport measurements, and to characterize subcellular transport processes in living tissues. At the same time these novel sensors may help to record structural rearrangements that occur when transporter bind their ligands or which occur during the transport cycle. We show here that, in contrast to the interpretation of results from crystallization studies (*Khademi et al., 2004*), AMTs and MEPs both change conformation during transport, and thus most probably do not function as 'passive gas channels'. The nature of the conformational changes, which here are monitored in ensembles, will require crystallization of the transporter in different states, as recently achieved for a variety of transporters (*Forrest and Rudnick, 2009*; *Shimamura et al., 2010*; *Jiang et al., 2012*; *Krishnamurthy and Gouaux, 2012*).

## AmTrac provides an activity proxy

To create sensors that report substrate binding, a circularly-permutated form of GFP (*Tian et al., 2009*) was inserted into the center of the cytosolic loop in AMT1;3 (and AMT1;2 and MEP2) at the pseudo-symmetry axis between the two antiparallel repeats of the transporter. The characteristics of the AmTrac fluorescence response, namely (i) the indistinguishable transport and fluorescence response kinetics, (ii) the reversibility, (iii) the tight link between transport activity for the FI response, (iv) the

**Table 1.** List of suppressor mutations, their location and frequency

| Mutation | Location | # | % |
|---|---|---|---|
| Q61E | TM 1 | 3 | 5 |
| Q61H | TM 1 | 5 | 9 |
| F64S | TM 1 | 3 | 5 |
| I140S | TM 3 | 1 | 2 |
| A141E | TM 3 | 26 | 46 |
| A142P | TM 3 | 13 | 23 |
| I150T | TM 3 | 2 | 4 |
| D464V | C-tail | 3 | 5 |
| Total | | 56 | 100 |

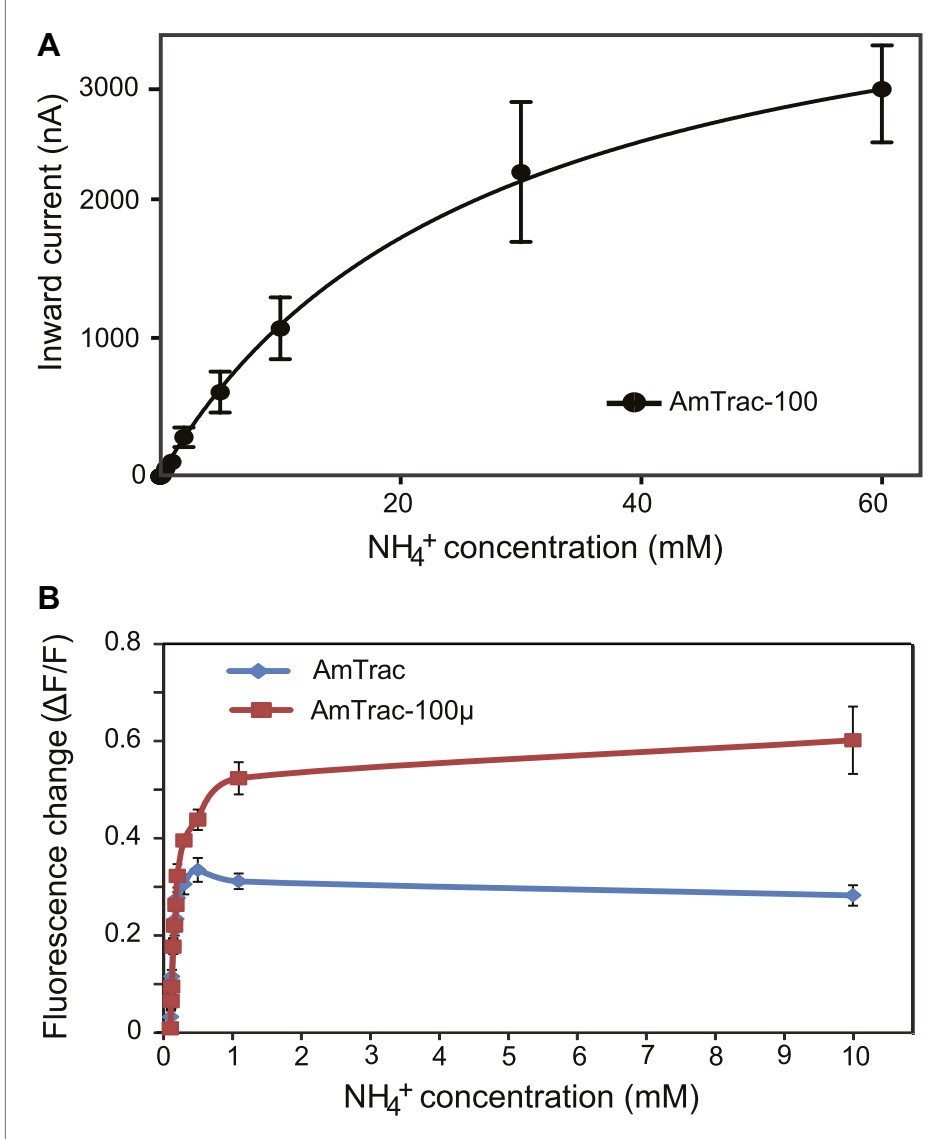

**Figure 8**. Characterization of a high capacity sensor variant. (**A**) Kinetics of $NH_4^+$-induced currents of AmTrac-100. The $Km$ was 3.4 ± 0.56 mM. The data were fitted to Michaelis–Menten kinetics. Oocytes were clamped at −120 mV (independent data from three different oocytes recorded from three different frogs). (**B**) Titration of the fluorescent response of AmTrac (blue line) and AmTrac-100µ (red line). Data are normalized to water-treated controls (0; mean ± SE: n = 3). AmTrac kinetics shown here for comparison are the same as **Figure 4C** and **Figure 5D**.

correlated shifts in affinity for ammonium transport and FI response, and (v) the restoration of the FI response by reconstitution of transport activity in suppressor mutants, strongly indicate that AmTrac sensors measure processes that occur during the transport cycle, or are intimately and quantitatively connected to transport activity. Since binding of the substrate to the transporter is part of the transport cycle, our data do not exclude the possibility that the sensors report extracellular binding of ammonium and therefore measure extracellular ammonium levels rather than activity.

## The role of substrate binding and the transport cycle for signaling through transceptors

AMTs have been suggested to function as both transporters and receptors for ammonium (**Lima et al., 2010**). Other transporters with a dual function as transporters and signaling proteins include UhpT from *E. coli* (**Schwöppe et al., 2003**), Pho84p and GAP1p from *Saccharomyces* (**Thevelein and**

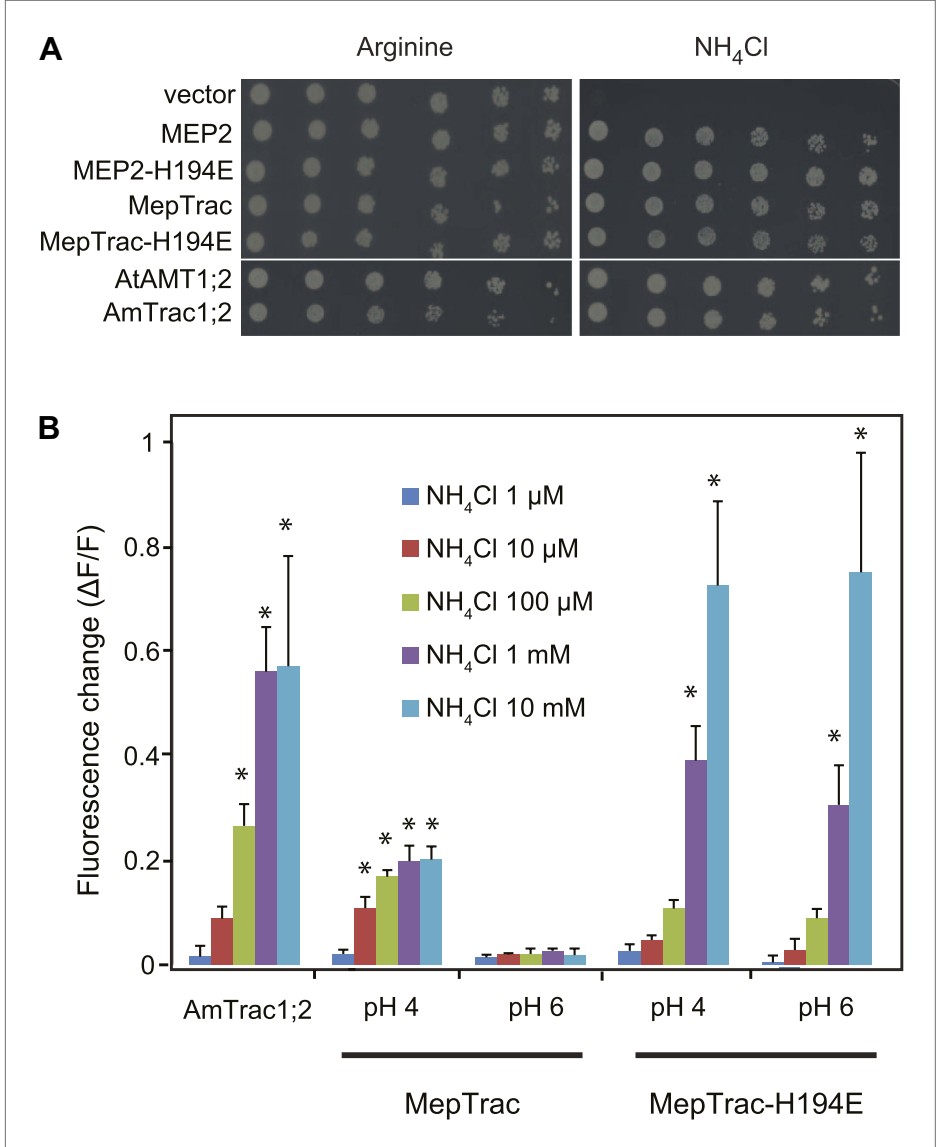

**Figure 9**. AmTrac1;2 and MepTrac response. (**A**) Growth complementation of *Δmep1,2,3* yeast expressing MEP2, MepTrac, their H194E mutants or AmTrac1;2 on solid media containing 2 mM $NH_4Cl$ or 1 mM arginine (growth control) as sole nitrogen source for 3 days. (**B**) Fluorescence response of *Δmep1,2,3* yeast expressing AmTrac1;2, MepTrac or MepTrac-H194E to $NH_4Cl$ at the indicated concentrations at pH 4 or 6. AmTrac1;2 was tested at pH 6. Data are normalized to water-treated control (0) (mean ± SE; n = 3; SNK test: *p<0.01).

*Voordeckers, 2009*) and Chl1 from Arabidopsis (*Ho et al., 2009*). The mechanism by which signaling is transmitted is unclear, specifically how substrate binding or conformational rearrangements during the transport cycle are necessary for triggering the receptor activity. Recent work by the Thevelein group has shed some light on the relative role of binding vs transport in this process. Some transporter-like receptors, such as the glucose transporter-like SNF3p and RGT1p from *S. cerevisiae*, appear to be unable to transport, suggesting that binding is sufficient for triggering a conformational rearrangement that is detected by intracellular signaling proteins (*Thevelein and Voordeckers, 2009*). The yeast Pho84 phosphate transceptor functions both in phosphate transport and mediates activation of the protein kinase A pathway. The finding that phosphate analogs that do not appear to be transported retain signaling activity is consistent with the hypothesis that a complete transport cycle is not required for signaling. However, further analyses suggested that ligand binding is insufficient for signaling, therefore the authors concluded the receptor activity is triggered by specific conformational rearrangements

that occur during a specific phase of the transport cycle (*Popova et al., 2010*). Mutations in the presumed proton-binding site severely reduced transport but not signaling, demonstrating that transport and signaling activities are separable (*Samyn et al., 2012*). Combined analysis of the action of amino acid analogs and mutagenesis of the putative binding site of the amino acid transceptor GAP1 on transport and signaling have indicated that, similar as in PHO84, signaling requires a ligand-induced conformational change that corresponds to a step in the transport cycle but does not appear to require a the full cycle (*Van Zeebroeck et al., 2009*). It is therefore conceivable that the fluorescent change observed in the fluorescent ammonium sensors presented here is due to either (i) ammonium binding, (ii) a specific set of steps in the transport cycle, or (iii) the full transport cycle of the transporter.

## Insights into the transport mechanism of AMTs

Crystal structures of AMTs from *Escherichia coli* (*Khademi et al., 2004*; *Zheng et al., 2004*) and *Archaeoglobus fulgidus* (*Andrade et al., 2005*) did not reveal obvious conformational differences when generated in the presence or absence of ammonium; observations that led to the proposition that AMTs are rigid gas channels (*Khademi et al., 2004*). Electrophysiological studies challenged the hypothesis that AMTs transport the uncharged form ammonia by demonstrating that plant AMTs are electrogenic and show minimal or no ammonia conductivity (*Mayer et al., 2006*). The transport of the charged form of ammonium is further supported by the finding that mutations in the pore release the transporter from a gating control, leading to massively increased transport capacity and an up to 100-fold increase in the $K_m$ (*Figure 8A*; *Loqué et al., 2009*).

Moreover, we provided evidence for the existence of open and closed states by demonstrating an allosteric feedback inhibition of transport activity by phosphorylation of residues in the *trans*-activating cytosolic C-terminus (*Loqué et al., 2007*; *Lanquar et al., 2009*). The fluorescence response of AmTrac variants and MepTrac supports the hypothesis that AMT1;3, AMT1;2 and MEP2 undergo conformational changes during transport. Our results are consistent with the hypothesis that TMH-V and -VI of AMT move relative to each other as a result of ammonium binding or during ammonium translocation, thus affecting the conformation of L5–6 and thereby gating the inserted mcpGFP.

## Potential to transfer the concept to other transporters, receptors, and enzymes

cpGFP has successfully been deployed to monitor conformational changes in a variety of proteins, for example, calmodulin (*Baird et al., 1999*; *Nagai et al., 2001*, *2004*; *Nakai et al., 2001*), metal ion binding domains (*Baird et al., 1999*; *Mizuno et al., 2007*), a periplasmic sugar binding protein (*Marvin et al., 2011*), PKG isoforms for measuring cGMP (*Nausch et al., 2008*), and a voltage sensing domain of a protein phosphatase (Ci-VSP) (*Gautam et al., 2009*). While none of the sensors measures activity, they report ensemble changes in conformational states of the respective proteins. Here we demonstrate that insertion of cpGFP can be more widely used to monitor activity states of proteins, specifically ammonium transporters.

Interestingly, the pseudo-symmetry of AMTs with an inverted repeat of a five TMH containing domain is comparable to that found in the LeuT transporter (*Krishnamurthy and Gouaux, 2012*). LeuT carries a substrate-binding site at the interface of the two repeats and undergoes a transport cycle for which multiple states have been identified (*Forrest and Rudnick, 2009*; *Rudnick, 2011*). Similarly, the transport pore of the AMTs is located between the two pseudo-symmetric halves, with TMH-V and -VI (connected by L5–6) carrying key residues for ammonium translocation. Our data are consistent with the occurrence of structural rearrangements during transport; crystal structures for different states will be required to determine the conformational state during transport.

## Conclusion

In summary, we developed an in vivo reporter of activity states for a set of ammonium transporters. The combination of two ammonium transport activity sensors from different organisms will be useful for dissecting endogenous control mechanisms, because heterologous proteins are not expected to be subject to the same regulatory control and can thus serve as references. As a next step, we will deploy AmTrac and MepTrac in both Arabidopsis roots and in yeast to study their regulation in vivo and to identify regulators. The measurement of flux through a transporter by reporting state changes, as shown here, might be applicable to other transporters or enzymes for monitoring in vivo fluxes, for

example, in the context of neurotransmission. Such in vivo flux analyses will aid in understanding the mode of action of proteins and may serve as tools for drug discovery.

## Materials and methods

### DNA constructs

All transporter and sensor constructs were inserted in the yeast expression vector pDRf1-GW, containing the f1 replication origin, GATEWAY cassette, PMA1 promoter fragment, ADH terminator, and the URA3 cassette for selection in yeast (*Loqué et al., 2007*). The 1494 bp ORF of AMT1;3 from Arabidopsis (At3g24300) was used as base for AmTrac construction.

The *XbaI* restriction site (tctaga) was inserted in different positions of AMT1;3 (after amino acids 233, 312, 364 and 448) using Kunkel mutagenesis (*Kunkel et al., 1991*).

mcpGFP was generated by amplifying the domains of EGFP corresponding to amino acids 150–239 and 1–144 with the primer pairs *EGFP-150-for/EGFP-239-rev* and *EGFP-1-for/EGFP-144-rev*, respectively (see *Supplementary file 1*).

The two amplified fragments were gel-purified by a commercial kit (Machery-Nagel, Düren, Germany), digested by *AgeI* (New England Biolabs, Ipswich, MA) and ligated by T4 DNA ligase (New England Biolabs). The resulting cpGFP, where the domains 150–239 and 1–144 were connected by the linker coding GGTGGS, was cloned into a pGEM-Teasy (Promega, Madison, WI). The additional internal mutations M154K, V164A, S176G, D181Y, T204V, A207K, V190I (positions referring to eGFP sequence), which had been shown to improve stability of cpGFP (*Akerboom et al., 2009*; *Tian et al., 2009*), were introduced by Kunkel mutagenesis (*Kunkel et al., 1991*) to generate mcpGFP. Finally, mcpGFP was amplified with the primers cpGFP-for and cpGFP-rev (*Supplementary file 1*). Similarly, mTFP and Venus were amplified with primers mTFP-for and mTFP-rev, and Venus-for Venus-rev, respectively (*Supplementary file 1*). The purified mcpGFP-, mTFP- and Venus-encoding fragments were digested by *XbaI* (New England Biolabs) and ligated into *XbaI* digested pDRf1-GW containing AMT1;3 versions that harbor the sequence corresponding to the *XbaI* restriction site in different cytosolic loops, to generate the fusion constructs AMT1;3-mcpGFP, AMT1;3-TFP and AMT1;3-Venus in positions 233, 312, 364 and 448 (*Figure 1B*).

To vary the linker regions between mcpGFP and AMT1;3 in position 233 (*Figure 2A*), we took advantage of homologous recombination in yeast between two DNA fragments sharing sequence homology. We co-transformed yeast with the pDR-AMT1;3 opened at position 233 by *XbaI* digestion, and the mcpGFP fragments amplified by PCR with the primer pair cpGFP-hom-for/rev containing the variable sequence encoding the linkers (*Supplementary file 1*). The amplification products contained mcpGFP flanked by the variable linker sequences and about 30 bp homologous to the region around the insertion point 233 of AMT1;3. The transformed yeast contained the pDR-AMT-mcpGFP vectors resulting from insertion of the ~800 bp mcpGFPs with linkers into the vector backbone, as confirmed by DNA sequencing.

Similarly, we used homologous recombination to generate random variants of the linker preceding mcpGFP (*Figure 5A*). For this case, mcpGFP was amplified with the primer pairs cGFP-hom-for-deg and cpGFP-hom-rev-FN (*Supplementary file 1*).

Homologous recombination was also used to insert mcpGFP in different positions along L5–6 (*Figure 3B*) and to generate deletions in the loop (*Figure 3C*). In this case, the 30-bp-long regions of the primers that overlapped the AMT1;3 sequence flanked the different insertion points (228–236) and contained the appropriate deletions.

Point mutations for inactivation of AMT1;3 and AmTrac were generated by Kunkel mutagenesis (D202N, G460D, T464D; *Figure 6*).

MepTrac was generated by overlapping PCR. The 1500 bp *S. cerevisiae* MEP2 cDNA was amplified in two separate halves (corresponding to amino acids 1–217 and 218–499) with the primers attB1-MEP2-for and MEP2-217-rev, and MEP2-218-for and attB2-MEP2-rev, respectively (see *Supplementary file 1*). Primers contained the MEP2-specific sequences plus the Gateway attB sites for subsequent BP cloning. A third fragment containing the mcpGFP with the linkers coding for the amino acids LS and FN (preceding and following the mcpGFP, respectively) was generated by amplifying the mcpGFP from the pDR-Amtrac-LS template with the primers cpGFP-LS-MEP2-for and cpGFP-LS-MEP2-rev (*Supplementary file 1*). The primers contained a region specific to mcpGFP, a sequence coding for the linkers and about 30 bp homologous to the region around the insertion point 217 of MEP2. The three

fragments were gel-purified by a commercial kit (Machery-Nagel) and 1 µl of each fragment was added to a PCR reaction tube (Phusion Taq; Finnzymes Thermo Fisher, Waltham, MA) containing 200 µM dNTPs and amplified for 10 cycles without addition of any primer. 1 µl of this reaction was further amplified for additional 30 cycles in a PCR reaction tube (Phusion Taq; Finnzymes) containing 200 µM dNTPs and the primers attB1-MEP2-for and attB2-MEP2-rev. The amplified fragment was gel-purified by a commercial kit (Machery-Nagel) and cloned into a pDON221 vector by Gateway BP reaction (Invitrogen Life Technology, Paisley, United Kingdom), following manufacturer's instructions. The resulting pENTRY-MepTrac construct was introduced into TOP10 competent cells (Invitrogen). The yeast vector harboring MepTrac was then created by Gateway LR reaction between pENTRY-MepTrac and pDRf1-GW, following manufacturer's instructions. Sequence of Meptrac in the resulting pDR-MepTrac was verified by sequencing. MepTrac-H194E was generated by single-point mutagenesis of MepTrac by a commercial kit (QuickChange; Stratagene Agilent, Santa Clara, CA).

AmTrac1;2 was generated by overlapping PCR by the same strategy used for MepTrac. The cassette of mcpGFP with LS/FN linkers was inserted into the *A. thaliana* AMT1;2 cDNA (1545 bp, At1g64780) after the codon for the amino acid R242, in the middle of predicted loop5–6. Primers are listed in *Supplementary file 1*.

For functional assays in *Xenopus* oocytes, the cDNAs of AMT1;3, AmTrac, AmTrac-LS, and AmTrac-100µ were cloned into the oocyte expression vector pOO2-GW (*Loqué et al., 2009*).

## Yeast culture

Yeast strains used in this study were 31019b [*mep1Δ mep2Δ::LEU2 mep3Δ::KanMX2 ura3*], a strain in which all three endogenous MEP ammonium transporter genes had been deleted, and its parental strain 23344c [*ura3*], considered here as wt (*Marini et al., 1997*). Yeast was transformed using the lithium acetate method (*Gietz et al., 1992*) and selected on solid YNB (minimal yeast medium without amino acids and without ammonium; Difco BD, Franklin Lakes, NJ) supplemented with 3% glucose and 1 mM arginine. Single colonies were inoculated in 5 ml liquid YNB supplemented with 3% glucose and 0.1% proline under agitation (230 rpm) at 30°C until $OD_{600nm}$=0.8. The liquid cultures were diluted $10^{-1}$, $10^{-2}$, $10^{-3}$, $10^{-4}$, $10^{-5}$ and $10^{-6}$ in water and 5 µl of each dilution was spotted on solid YNB medium buffered with 50 mM MES/Tris, pH 5.2 and supplemented with 3% glucose and either $NH_4Cl$, $(NH_4)_2SO_4$ or 1 mM arginine as the sole nitrogen source. After 3 days of incubation at 30°C, cell growth was documented by flatbed scanning the plate at 300 dpi in grayscale mode. For quantification of yeast growth (*Figure 7E*), the pixel intensity of each spot (here $10^{-2}$ dilution) were quantified using ImageJ software, and normalized to AmTrac spot intensity (100%). Quantification of spot intensity of $10^{-1}$ and $10^{-3}$ dilutions gave similar results (not shown).

## Fluorimetric analyses

Fluorimetric analyses are described in detail at Bio-protocol (*Ast et al., 2015*). Yeast cultures were washed twice in 50 mM MES buffer, pH 6.0, and resuspended to $OD_{600nm}$=0.5 in MES buffer supplemented with 5% glycerol to delay cell sedimentation. For MepTrac and MepTrac-H194E measurements, cells were washed twice in purified water and resuspended in 25 mM sodium citrate buffer, pH 4.0 or 6.0, supplemented with 5% glycerol. Fluorescence was measured by a fluorescence plate reader (Safire; Tecan, Männedorf, Germany), in bottom reading mode using a 7.5 nm bandwidth for both excitation and emission (*Bermejo et al., 2010, 2011*). To measure fluorescence response to substrate addition, 50 µl of substrate (dissolved in water as 500% stock solution) were added to 200 µl of cells in a 96-well plate (Greiner, Monroe, NC). Fluorescence was measured as emission at $\lambda_{em}$ 513 nm using excitation at $\lambda_{exc}$ 488 nm. Response data are presented as $(F_{water} - F_{treatment})/F_{treatment}$.

## Photophysical characterization

Steady-state emission measurements were performed using a Fluoromax-P fluorescence spectrometer (Horiba Jobin Yvon, Kyoto, Japan) in 3.5-ml silica cuvettes (Hellma Analytics, Müllheim, Germany). 2 ml liquid yeast cultures were supplemented with 0.5 ml $NH_4Cl$ (10 mM, 1 mM, 100 µM or 10 µM) or water as control. The samples were allowed to incubate for 20 min. The excitation and emission spectra were recorded at $\lambda_{em}$ 514 nm and $\lambda_{exc}$ 485 nm, respectively, with a step size of 1 nm and five repeats were taken for averaging. Untransformed yeast cells served as blank.

## Fluorescence microscopy

Confocal sections of yeast cells expressing the sensors (*Figure 6A*) were acquired on an inverted confocal laser scanning microscope (SP5; Leica, Wetzlar, Germany). To record fluorescence intensities

in single cells over time, yeast cells were trapped as a single cell layer in a microfluidic perfusion system (Y04C plate; Onyx, Cellasic, Hayward, CA) and perfused with either 50 mM MES buffer, pH 6.0, or buffer supplemented with $NH_4Cl$ (*Bermejo et al., 2010*, *2011*). Cells were imaged on a spinning disk confocal microscope (Yokogawa CSU-X1, Mitaka, Japan; Leica DMI6000) equipped with a motorized stage (ASI). Fluorescence was excited by a solid state laser at 488 nm; emission was detected using a 525/50 nm filter set (Semrock, Rochester, NY) and an electron multiplying charge coupled device (EMCCD) camera (Evolve; Photometrics, Tucson, AZ). Measurements were taken every 2 min, with 100 ms exposure time using Slidebook 5.0 image acquisition software (Intelligent Imaging Innovations, Denver, CO). To account for lateral shift during imaging, the image stacks were post-registered using the StackReg plugin for ImageJ (*Thevenaz et al., 1998*). Fluorescence pixel intensity was quantified using Fiji software; single cells were selected and analyzed with the help of the ROI manager tool.

## Functional expression of AMT1;3 and AmTracs in *Xenopus* oocytes

For in vitro transcription, pOO2-AMT1;3 and AmTracs were linearized with *MluI*. Capped cRNA was transcribed in vitro by SP6 RNA polymerase using mMESSAGE mMACHINE kits (Ambion, Austin, TX). *Xenopus laevis* oocytes were obtained by surgery, manually dissected, and defolliculated with collagenase (Sigma, St. Louis, MO). The oocytes were injected by Roboinjector (Multi Channel Systems, Reutlingen, Germany; (*Pehl et al., 2004*; *Leisgen et al., 2007*)) with distilled water (50 nl as control), AMT1;3 cRNA (50 ng in 50 nl), AmTrac cRNA (50 ng in 50 nl), AmTrac-LS cRNA (50 ng in 50 nl), or AmTrac-100μ cRNA (50 ng in 50 nl). Cells were kept at 16°C in ND96 medium containing 96 mM NaCl, 2 mM KCl, 1.8 mM $CaCl_2$, 1 mM $MgCl_2$, and 5 mM HEPES, pH 7.4, containing gentamycin (50 μg/μl).

## Electrophysiological measurements

Measurements were made in a solution as previously described (*Loqué et al., 2009*). Oocytes were voltage-clamped at −120 mV and measured by two-electrode voltage-clamp (TEVC) Roboocyte system (Multi Channel Systems) (*Pehl et al., 2004*; *Leisgen et al., 2007*).

## Statistical analyses

All experiments were performed in triplicate, unless specified otherwise. Reported values represent mean and standard deviation. The effects of treatments on the fluorescence response were compared using analysis of variance (ANOVA): the factors (sample, treatment) were treated as fixed factors. Prior to analyses, homogeneity of the variances was tested by Cochran's test. Whenever the ANOVA revealed significant differences among treatments, post hoc comparisons were performed using the SNK test. ANOVAs were performed using the GMAV5 software package (University of Sydney, Australia).

# Acknowledgements

We are very grateful to Loren Looger for advice regarding linker composition for insertion of the mcpGFP in the fusion construct, and to David Ehrhardt for critical reading. CA was supported by Carnegie Institution and DAAD. SLAA acknowledges financial support by the DFG (AN 676/1). We are very grateful for support provided by HG Löhmannsröben (University of Potsdam), Niko Hildebrandt (Université Paris-Sud), Joost van Dongen (MPI of Molecular Plant Physiology). We would like to thank Miriam Goodman (Stanford University) for providing Xenopus oocytes.

# Additional information

### Funding

| Funder | Grant reference number | Author |
| --- | --- | --- |
| National Science Foundation | MCB-1021677 | Wolf B Frommer |
| DOE BER Office of Science | DE-AC02-05CH11231 | Dominique Loqué |
| Deutsche Forschungsgemeinschaft | AN 676/1 | Susana LA Andrade |

The funders had no role in study design, data collection and interpretation, or the decision to submit the work for publication.

## Author contributions

RD, CA, Conception and design, Acquisition of data, Analysis and interpretation of data, Drafting or revising the article; DL, GG, Conception and design, Analysis and interpretation of data; C-HH, VL, Acquisition of data, Analysis and interpretation of data; SG, Analysis and interpretation of data; SLAA, WBF, Conception and design, Analysis and interpretation of data, Drafting or revising the article; MUK, Analysis and interpretation of data, Drafting or revising the article

## Additional files

### Supplementary files

• Supplementary file 1. Primers used for AmTrac and MepTrac cloning.

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
