## [Decision Letter]

Thank you for sending your work entitled “Fluorescent sensors for activity states of ammonium transceptors in live cells” for consideration at *eLife*. Your article has been favorably evaluated by a Senior editor, Detlef Weigel, a Reviewing editor, and 2 reviewers.

The following individuals responsible for the peer review of your submission want to reveal their identity: Richard Aldrich (Reviewing editor), and Christopher Miller and Jonathan Javitch (peer reviewers).

The Reviewing editor and the two reviewers discussed their comments before we reached this decision, and the Reviewing editor has assembled the following comments to help you prepare a revised submission.

The manuscript describes the development, refinement, and optimization of a set of GFP-based fluorescent reporters for a transport-active state of an AMT-family ammonium transporter from Arabidopsis. The experiments convincingly show that the fluorescent intensity of these reporters – circularly permuted GFP constructs fused to AMT1;3 – changes according to the level of extracellular NH_4_^+^ in the 0.1 mM range, and that this signal nicely tracks the NH_4_^+^ transport activity as measured directly by voltage-clamp methods in oocytes heterologously expressing the transporter. Thus, the authors argue, we now have a convenient, quantitative reporter for AMT transport activity that may be used in live-cell imaging. Furthermore, they point out, NH_4_^+^ transport is generally inaccessible to direct measurement, since there is no radioactive isotope or other standard means of following uptake. Since NH_4_^+^ is a key substrate in plant physiology, these reporters represent an important technical advance. Despite the fact that the reviewers are not convinced that the sensors report transport as opposed to binding, they feel that the approach and the ability to report NH_4_^+^ concentration represent an important advance. The following changes will be necessary for acceptance:

The paper must clearly state and explain that the sensors report transport activity *or* ammonium binding (or both).

The major concern is the lack of discussion or discrimination of the possibility that the sensor was registering ammonium binding and not ammonium transport per se. Many of the features used to argue that the fluorescence intensity change requires transport do not seem to differentiate ammonium binding and transport – a concentration dependence and reversibility, for example.

The work is convincing, satisfying, and carefully done. While we are convinced that these reporters can be used to track extracellular NH_4_^+^ concentration seen by AMT transporters in these plant cell membranes, we are not at all convinced that the signal reports transport activity, which the authors repeatedly conflate with substrate binding. On the other hand, they may be correct in equating the two for the following reason. AMT 1;3 is thought to act in a dual-function manner – as a transporter for NH_4_^+^, but also as a signaling protein for extracellular NH_4_^+^. It has therefore been dubbed a “transceptor”, an evocative term familiar to plant aficionados but not to the general reader. The protein, in its “receptor” mode, is therefore thought to undergo a conformational change upon NH_4_^+^ binding, leading to signal transduction; this is the conformational change the reporters are presumably reporting. But this need not be equivalent to transmembrane movement of substrate, which necessarily involves a steady-state cycling of the protein through multiple conformations (inward-facing, empty – inward-facing, substrate bound, outward-facing, substrate bound, inward occluded, etc.). In that case, the reporter would be reporting a kind of average of these multiple states, not of the turnover itself, and certainly would carry no information about the rate of substrate turnover.

The authors write:

“Together these data strongly support the hypothesis that AmTrac measures ammonium concentrations and/or reports conformational state changes of the transporter.”

This is the one place where the authors seem to entertain the possibility of binding (ammonium concentrations on the outside), but the rest of the text assumes transport as the explanation. Perhaps the suppressor screen experiments argue for actual transport, but this would require more information on binding and how this might be differentiated in the mutants. This should be discussed. If the authors have data or can argue clearly for transport, the case should be made. Otherwise this needs to be discussed in the context of binding or transport. This seems particularly important since the change in GFP that is proposed to account for the change in fluorescence intensity seems to be a protonation change and not necessarily a conformational change (although of course a conformational change might be present).

The authors also make an effort to glean some mechanistic insight, but the inference that the channels are not passive and must change conformation requires the assumption that only a conformational change could underlie the intensity change, when in fact some sort of change upon binding might lead to a change in protonation without much of a conformational change.

These concerns are not seen as a deficiency of the reporters, which may be quite valuable in studies of ammonium handling in these plants. But the authors have failed to spell out the logic here, beyond using the word “transceptor” in the Introduction, and so leave the reader confused about what exactly is being reported by these fluorophores.

---

## [Author Response]

The key criticism was that we did not discuss the possibility adequately that our sensors report extracellular binding of ammonium. It is correct that our data do not allow us to discriminate between binding and conformational rearrangements in the transport cycle. We would argue that the strict correlation of fluorescence response and ability to transport, even in quantitative terms, supports the fact that we see activity, but since the two are expected to be coupled, we have here not separated them. We had actually left out a whole discussion of this aspect, and we had left out important data from studies performed by Thevelein on the role of binding versus transport in his “transceptors”. While we have currently no reason to believe that what we detect is the “receptor function” in our proteins, the discussion on how they determined the role of binding versus transport cycle for the signaling function is very informative. We have therefore added a paragraph in the Discussion that addresses this (“The role of substrate binding and the transport cycle for signaling through transceptors”):

“AMTs have been suggested to function as both transporters and receptors for ammonium (Lima, 2010). Other transporters that may have a dual function as transporters and signaling proteins include UhpT from *E. coli* (Schwöppe, 2003), Pho84p and GAP1p from *Saccharomyces* (Thevelein, 2009) and Chl1 from Arabidopsis (Ho, 2009). The mechanism of the receptor function is unclear, specifically whether substrate binding or conformational rearrangements during the transport cycle are necessary for triggering the receptor activity. Recent work by the Thevelein group has shed some light on the relative role of binding versus transport in this process. Some transporter-like receptors, such as the glucose transporter-like SNF3p and RGT1p from *Saccharomyces cerevisiae*, appear to be unable to transport, suggesting that binding is sufficient for triggering a conformational rearrangement that is detected by intracellular signaling proteins (Thevelein, 2009). The yeast Pho84 phosphate transceptor functions both transport as phosphate transporter and mediates activation of the protein kinase A pathway. The use of phosphate analogs which do not appear to be transported retain signaling activity, consistent with the hypothesis that for signaling to occur, a complete transport cycle is not required. However, further analyses suggested that ligand binding is insufficient for signaling, therefore the authors concluded the receptor activity is triggered by specific conformational rearrangements that occur during a specific phase of the transport cycle (Popova, 2010). Mutations in the presumed proton-binding site severely reduced transport but not signaling, demonstrating that transport and signaling activities are separable (Samyn, 2012). Combined analysis of the action of amino acid analogs and mutagenesis of the putative binding site of the amino acid transceptor GAP1 on transport and signaling have indicated that, similar as in PHO84, signaling requires a ligand-induced conformational change that corresponds to a step in the transport cycle but may not the full cycle (Van Zeebroeck, 2009).”

It is therefore conceivable that the fluorescent change observed in the fluorescent ammonium sensors presented here is due to either (i) ammonium binding; (ii) a specific set of steps in the transport cycle; or, (iii) the full transport cycle of the transporter.

*The paper must clearly state and explain that the sensors report transport activity* or *ammonium binding (or both)*.

We agree with the reviewers that we cannot differentiate between the two (three) possibilities. We have thus added a section to the discussion that addresses this question.

*The work is convincing, satisfying, and carefully done. While we are convinced that these reporters can be used to track extracellular NH*_*4*_^*+*^
*concentration seen by AMT transporters in these plant cell membranes, we are not at all convinced that the signal reports transport activity, which the authors repeatedly conflate with substrate binding. On the other hand, they may be correct in equating the two for the following reason. AMT 1;3 is thought to act in a dual-function manner – as a transporter for NH*_*4*_^*+*^*, but also as a signaling protein for extracellular NH*_*4*_^*+*^*. It has therefore been dubbed a “transceptor”, an evocative term familiar to plant aficionados but not to the general reader*.

The term is a contraction from transporter and receptor by Johan Thevelein for the yeast general amino acid permease Gap1p. Since then, it has been widely used by researchers in the fungal and plant communities for a wide spectrum of proteins with dual function. We have clarified this in the revised Introduction.

*The protein, in its “receptor” mode, is therefore thought to undergo a conformational change upon NH*_*4*_^*+*^
*binding, leading to signal transduction; this is the conformational change the reporters are presumably reporting. But this need not be equivalent to transmembrane movement of substrate, which necessarily involves a steady-state cycling of the protein through multiple conformations (inward-facing, empty – inward-facing, substrate bound, outward-facing, substrate bound, inward occluded, etc.). In that case, the reporter would be reporting a kind of average of these multiple states, not of the turnover itself, and certainly would carry no information about the rate of substrate turnover*.

We currently have no reason to believe that the conformational change that underlies (most likely) the fluorescence output is related to the “receptor” function of the transporter. No one, so far, has been able to separate transport and receptor functions. Moreover, AMT1;2 has, at least for now, not been shown to be a “transceptor”. But we fully agree with the Reviewers that we have no data, beyond our tight correlation between fluorescence output and transport activity (even quantitative in some cases), that clearly show whether the effect is caused by binding or a specific conversion during a predicted transport cycle or requires the full transport cycle. And we only see two states; whether there are additional states remains to be shown. We therefore followed the reviewers’ advice and cautioned statements throughout, and added a full paragraph to the Discussion that discusses this question, specifically in the context of “transceptors”, for which Thevelein has over the past years provided very interesting data regarding the binding versus transport cycle question. We apologize for having omitted this important discussion, and we are grateful for the opportunity to address this in the revised manuscript.

*The authors write*:

*“Together these data strongly support the hypothesis that AmTrac measures ammonium concentrations and/or reports conformational state changes of the transporter.*”

*This is the one place where the authors seem to entertain the possibility of binding (ammonium concentrations on the outside), but the rest of the text assumes transport as the explanation. Perhaps the suppressor screen experiments argue for actual transport, but this would require more information on binding and how this might be differentiated in the mutants. This should be discussed. If the authors have data or can argue clearly for transport, the case should be made. Otherwise this needs to be discussed in the context of binding or transport. This seems particularly important since the change in GFP that is proposed to account for the change in fluorescence intensity seems to be a protonation change and not necessarily a conformational change (although of course a conformational change might be present)*.

At present we have no data that support either hypothesis – whether the fluorescent change is caused by protonation (one may hypothesize that the conformational change in the transport leads to gating of the opening in the cage and thus affects protonation), or whether the transporter tweaks the fluorophore.

*The authors also make an effort to glean some mechanistic insight, but the inference that the channels are not passive and must change conformation requires the assumption that only a conformational change could underlie the intensity change, when in fact some sort of change upon binding might lead to a change in protonation without much of a conformational change*.

As far as we can tell, despite attempts to gain insights into cpGFP sensors, the mechanism is unclear, and our data do not provide new insights into the actual mechanism – tweaking of the fluorophore cage could affect protonation, gating of the hole in the cage could affect protonation, and there may be other potential explanations. In the case of Gcamp5, apparently dimerization plays a role. So we tried to stay away from speculation. We would also like to mention that the cpGFP is inserted on the cytosolic side of the membrane; thus it will not be affected by extracellular ammonium. Therefore, as the reviewers suggest, even if protonation is the direct cause of response, it still may need a conformational change (triggered by either binding or transport).

*These concerns are not seen as a deficiency of the reporters, which may be quite valuable in studies of ammonium handling in these plants. But the authors have failed to spell out the logic here, beyond using the word “transceptor” in the Introduction, and so leave the reader confused about what exactly is being reported by these fluorophores*.

At present, we hypothesize that our approach can be used for other transporters and does not necessarily depend on the use of a transceptor. To reiterate, the term “transceptor” was first used by Johan Thevelein in the context of the yeast Gap1P general amino acid permease, and then extended to the Pho84 phosphate transporter of yeast, as well as the plant nitrate transporter Chl1 (and now is widely used in both fields). A classical protein acting as transporter and receptor is the UDP-glucose sensor UhpC. The term simply contracts transporter and receptor into one word, reflecting the dual function of these proteins. The simplest hypothesis is that conformational changes during the transport cycle are monitored by intracellular proteins and these changes are used as information for triggering signaling cascades. Whether this is binding in the outward open state, a set of steps in the transport cycle, or the full cycle is currently being investigated. Data for Gap1p and Pho84 support the notion that the full transport cycle is not required, but that binding is insufficient. Whether AmTrac and MepTrac provide a fluorescence change upon ammonium addition because it might also function as a transceptor is unclear.

It will be interesting to test whether inactivation of by a regulatory protein (e.g., a kinase that phosphorylates T460) will block the fluorescent response. This is likely, given that a phosphomimic mutation leads to a loss of the fluorescent response. If this is the case, one could argue that AmTrac is an activity sensor.

So far, we have not been able to decouple the fluorescent response from transport activity, but it will be interesting to search for mutations that retain the fluorescent response but are transport-inactive.